# Systemic Bevacizumab for Recurrent Respiratory Papillomatosis: A Scoping Review from 2009 to 2022

**DOI:** 10.3390/children10010054

**Published:** 2022-12-26

**Authors:** Laura Torres-Canchala, Daniela Cleves-Luna, Oriana Arias-Valderrama, Estephania Candelo, María Angelica Guerra, Harry Pachajoa, Manuela Olaya

**Affiliations:** 1Fundación Valle del Lili, Centro de Investigaciones Clínicas, Cali 760032, Colombia; 2Departamento Medicina, Facultad de Ciencias de la Salud, Universidad Icesi, Cali 760031, Colombia; 3Centro de Investigaciones de Enfermedades Raras y Malformaciones Congénitas, Universidad Icesi, Cali 760031, Colombia

**Keywords:** adjuvant therapy child, adult, adult-onset recurrent respiratory papillomatosis juvenile-onset papillomatosis, recurrent respiratory papillomatosis, systemic bevacizumab, treatment

## Abstract

Background: Respiratory recurrent papillomatosis (RRP) is a fatal disease with no known cure. In severe RRP cases, systemic bevacizumab (SB) could be used as adjuvant therapy. Objective: This study aims to determine the extent and type of evidence in relation to the clinical outcomes of RRP after SB treatment. Methods: Participants with RRP of all genders are included in this scoping review. There were no exclusion criteria (country, language, or document type). The information sources included experimental, quasi-experimental, and analytical observational studies. Unpublished data will not be covered, but gray literature was covered. Screening, paper selection, and data extraction were all done by two independent reviewers. This procedure was performed blindly. Results: Of the 175 unique records found, 15 were eligible for inclusion. Fourteen studies were included after applying inclusion and exclusion criteria. Thirty-four patients in these studies came from the United States, India, Germany, Colombia, Argentina, Chile, and Spain. In total, 17 and 34 patients were below 18 years old and were adults respectively. The most commonly reported dose was 10 mg/kg, which was received by 25 (73.5%) patients. According to reports, 58.8% of patients completed the questionnaire. Twelve (35%) patients did not require a repeat surgery. The time interval between surgical procedures has increased for patients who require them. Conclusion: SB may be a promissory treatment and control option for RRP. More research is needed to evaluate the efficiency and adverse effects in various populations.

## 1. Introduction

Recurrent respiratory papillomatosis (RRP) and recurrent cutaneous papillomatosis (RCP) are devastating diseases with a high risk of long-term complications that impact patients’ and their families’ quality of life. The juvenile-onset RRP (JO-RRP) incidence rate is 4.3 per 100,000, whereas the adult-onset RRP (AO-RRP) incidence rate is 1.8 per 100,000 [1]. The two types of human papillomavirus (HPV) that cause RRP and RCP are HPV 6 and HPV 11. RRP and RCP are distinguished by the recurring growth of papillomas in the respiratory and gastrointestinal tracts, which can obstruct them in severe cases. There is no treatment for systemic papillomatosis [2].

Extra laryngeal papillomatosis has been seen in up to 30% of cases, with the oral cavity, trachea, and bronchi being the most common sites [3,4]. These patients have recurrent bronchiectasis, pneumonia, and decreased lung function [3,4]. Additionally, RRP is associated with a malignancy of RR: 8.0 (95% CI, 1.1–60.3) [3,4,5]. The age of disease onset has been reported as early as the first day of life, with a prevalence ranging from 1.45 to 2.93 per 100,000 children, with the disease being more aggressive at the earlier age of presentation [6]. When RRP appears in children under the age of 3, they are 3.6 times more likely to require surgical intervention [7].

Recurrent respiratory papillomatosis is difficult to manage. The current standard of care for laryngeal and gastrointestinal involvement is surgery to remove the papilloma while maintaining normal structure [8]. Methods, such as cauterization and laser therapy, have been used [8]. Nonetheless, due to severe side effects, such as respiratory tract and laryngeal scarring with posterior stenosis, microdebriders have taken its place. Despite the use of microdebriders to reduce tissue damage, surgical procedures in RRP are performed frequently, increasing the incidence of scar formation [8]. To avoid this, adjuvant therapy has been proposed.

The first line of adjuvants is antiviral management, but the results are heterogeneous and, in some cases, ineffective [9]. Monoclonal antibodies, such as anti-PDL-1 and bevacizumab, are another treatment option [10]. Bevacizumab is a monoclonal antibody that inhibits angiogenesis and tumor growth by targeting vascular endothelial growth factor [11]. This drug has been used as an adjuvant to surgical management in sub-lesional areas of RCP and RRP. Some authors have reported that high doses of sub-lesional bevacizumab could increase the time interval between procedures without adverse events but response and remission outcomes vary [12,13].

Despite the lack of clinical trials demonstrating the efficacy of systemic bevacizumab (SB) in recurrent papillomatosis, several case reports have demonstrated promising results with this therapy [14,15,16]. Morh et al. (2015) proposed SB as treatment option for RRP with rapid response in 2014 [17]. Because there are few reported cases of RRP treated with SB in the literature, there is insufficient information on dosage, effectiveness, and safety. Therefore, its use remains contentious [17]. Most common side effects are proteinuria, hypertension, respiratory tract hemorrhage, gastrointestinal hemorrhage, thromboembolic events, and wound complications [18].

## 2. Materials and Methods 

### 2.1. Search Strategy

The current scoping Review was performed following the framework proposed by Arksey and O’Malley, 2005 [19]. The search strategy was to answer the research question of the outcomes in patients with recurrent respiratory papillomatosis after systemic bevacizumab treatment. Search terms were defined that described two key concepts: (1) bevacizumab and (2) Recurrent Respiratory Papillomatosis; these terms were combined using Boolean operators OR (within critical constructed concepts) and AND (between key concepts). The search was conducted in the following databases: ClinicalTrials, EMBASE, MEDLINE (PubMed), LILACS, and Global Health (Ovid) (Appendix A).

### 2.2. Study Selection and Data Collection

Three individuals (L.T., D.C. and O.A.) independently screened all titles and abstracts generated by the searches. The same investigators who determined final eligibility reviewed the articles selected for full-text review. A third independent reviewer (M.O.) resolved disagreements over article eligibility.

### 2.3. Eligibility Criteria

All studies with a report, description, or analysis of SB in RRP were included. Exclusion criteria were as follows: (a) no full-text available, (b) administration route other than intravenous bevacizumab, (c) reviews, and (d) other therapy other than bevacizumab.

### 2.4. Study Selection and Data Abstraction

We constructed a standardized data extraction grid adapted from the table made by Best et al. [17]. Two independent investigators (L.T. and D.C.) extracted the following data: study type, author, year of publication, patient recurrent respiratory papillomatosis characteristics, management used before SB, and outcomes.

### 2.5. Statistical Analysis

If the data were normally distributed, dichotomous variables were reported as *n* (%) and continuous data as the median (IQR) or mean (*SD*). The analyses were performed using Stata^®^ 14.0 (Stata Corp., 2014, College Station, TX, USA). The case report was approved by the institutional IRB.

## 3. Results

This research strategy yielded 175 distinct records. This scoping study include up to 15 studies. Following the application of inclusion and exclusion criteria, 14 studies were included. Figure 1 shows the flowchart diagram for study selection.

Table 1 summarizes the clinical characteristics of patients with RRP before SB treatment. The studies included were published between 2009 and 2021. A total of 34 patients were described in these studies from the United States (three case reports; *n* = 3, four case series; *n* = 12), India (one case report; *n* = 1), Germany (one case report; *n* = 1, one case series; *n* = 5), Colombia (one case series; *n* = 3), Argentina (one case report; *n* = 1), Chile (one case report; *n* = 1), and Spain (one case series; *n* = 2). In total, In total, 17 and 34 patients were under 18 years old and were adults, respectively. The median age for those under 18 years old was 8 yr (min = 0.58 and max = 16), and the median for adults was 34 years (min = 18 and max = 87). There were 23 males (67.6%).

Twenty-six of the 34 patients had laryngeal lesions, 19 had tracheal lesions, and 17 had lung lesions. Twelve patients had tracheal, laryngeal, and lung lesions. Before bevacizumab therapy, 24 of the 34 patients had undergone surgical procedures. The reported interventions had a median of 13 interventions (min = 3 interventions and max = 500 interventions).

Table 2 summarizes SB treatment, dosage, and outcomes in patients with RRP. Various doses were reported at various schemes. The most commonly reported SB dose was 10 mg/kg, which was received by 25 (73.5%) patients. The initial dose intervals with this dose ranged from 2 to 5 weeks. Two patients received 5 mg/kg with a 2-week interval between doses, and one received the same dose but with a 3-week interval. Two patients were given 15 mg/kg at 3-week intervals. The authors reported an initial dose of 5 mg/kg in four patients, followed by one increase to 7.5 mg/kg and two increases to 10 mg/kg.

A stable dosing interval was reported in 30/34 patients. The median regular dosing interval in these patients was 8 weeks (min = 3 weeks and max = 16 weeks). This interval was reported to be between 19 and 83 days in one patient. When the papers were written, most patients were still receiving SB at regular dosing intervals. Twenty patients (58.8%) had complete responses. Eight of them responded extremely well. Twelve patients responded partially.

Due to RRP, 12 (35%) patients did not require surgery after receiving SB. Six of the patients required up to two additional interventions due to punctual lesions that did not respond. Seven microdebrider procedures were required for one patient. Seven patients required surgical interventions, but the time between them was increased from weeks to months. Due to malignant transformation, one patient required a laryngectomy.

## 4. Discussion

Respiratory recurrent papillomatosis is considered a medical challenge. First, patients’ quality of life is significantly reduced due to the constant growth of lesions that require frequent surgical management, by either cold or laser surgery. Second, because lung tissue spreads easily, surgical access becomes more difficult. Third, these lesions have the potential to progress to cancer [7]. There is currently no cure for this pathology, so therapy focuses on managing and controlling lesions that appear gradually, though more quickly in JO-RRP than in AO-RRP. SB has been proposed as an adjuvant therapy for RRP to reduce the frequency of surgical interventions. Although some case reports show therapeutic success, no clinical trials support the use of SB in non-malignant pathologies [30]. In this scoping review, we found 34 cases of patients with RRP treated with SB described in the literature, including our patient, the youngest boy with cutaneous RRP who has received SB. SB may be an effective adjuvant therapy for patients with severe RRP.

In the pediatric population, RRP is the most common benign neoplasm. Although the larynx is the most commonly affected site, it can also affect the respiratory and gastrointestinal tracts, has a high capacity for dissemination and recurrence despite treatment, and exhibits unpredictable behavior. RRP increases the risk of obstruction, requiring multiple surgical interventions that impact the quality of life for patients and their families [13]. Because of the high risk of recurrence, JO-RRP is more aggressive than AO-RRP, particularly if RRP is diagnosed before 3 years [8].

Adjuvant therapy aims to extend the time between operations or reduce the recurrence of RRP lesions [8]. Interferon α or cidofovir, as well as a topical treatment, may be used to extend the time between surgical procedures, but there is no good evidence of their efficacy [8]. The American Society of Pediatric Otorhinolaryngology studied a large group of patients with RRP; 21% received adjuvant therapy with cidofovir or interferon α. Despite treatment, the patients’ lesions progressed to the trachea, bronchus, and lung parenchyma [13].

SB was described as an effective adjuvant therapy for RRP lesions in this scoping review that included 34 patients with severe RRP described in 14 articles. After treatment, 14 (41.2%) of these patients had a complete response. Twelve (35.3%) patients did not require surgical intervention. Surgical interventions were drastically reduced in patients who required them. This adjuvant therapy appears to be a promising option in patients with RRP. More research is needed to determine the efficacy of SB in this population.

### 4.1. Side Effects

In the pediatric population, the only side effects observed with SB treatment were mild proteinuria in two patients, with no long-term side effects. these effects did not modify the treatment of the patients.

### 4.2. Limitations

This scoping review has some limitations, and its interpretation should be based on its characteristics and the conditions under which it was conducted. First, we included information from case reports, case series, and, in one article, responses to a survey of several experts who have used SB in their patients, making it susceptible to publication and information bias. Second, due to the nature of the articles, we did not conduct quality assessments on the included studies. Third, the population includes people of all ages, with varying RRP presentations, making it impossible to extrapolate bevacizumab indications to specific individuals. Fourth, because there is little research on this topic, there is still some uncertainty about the dosage, initial and standard administration intervals, and long-term adverse events. Thus, we are unable to make firm recommendations regarding the use of SB in RRP. Nonetheless, given the available data, SB could be a promising adjuvant therapy in RRP, with a likely high impact on patients’ quality of life, so this scoping review serves as a call to conduct additional research on this topic.

## 5. Conclusions

SB may be a viable option for extending the duration of RRP remission and improving the quality of life for patients and their families. However, more research is needed to evaluate the efficacy and adverse effects in various populations.

## Figures and Tables

**Figure 1 children-10-00054-f001:**
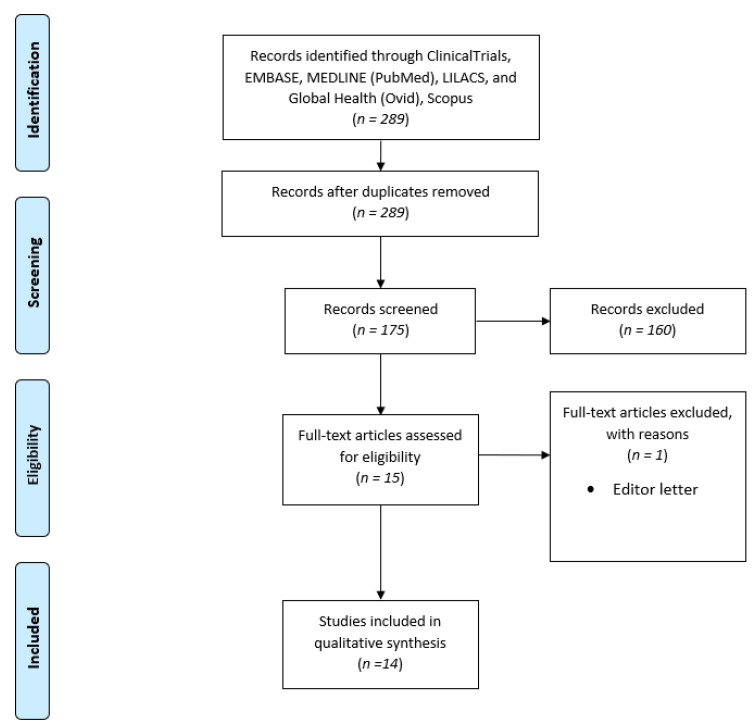
PRISMA Flowchart.

**Table 1 children-10-00054-t001:** Clinical characteristics of patients with RRP prior systemic bevacizumab treatment. CR, Complete Regression, PR, Partial Regression, VGPR, very good partial remission, NA Not Applicable.

Study	Author	Country	Year	Age at Treatment Start	Gender	Age at Onset	Larynx	Trachea	Lung	Number of Surgical Interventions Needed	Surgical Intervals	Prior Therapy
1	Best et al. [17]	United States	2017	20	F	Juvenile	Y	Y	Y	--	3 weeks	Local: Cidofovir, bevacizumab, photodynamic therapy. Systemic. Interferon alpha
2	Best et al. [17]	United States	2017	12	F	Juvenile	Y	Y	Y	--	1–4 weeks	Local: Cidofovir Systemic: Interferon alpha, propanolol, celecoxib, Gardasil
3	Best et al. [17]	United States	2017	16	F	Juvenile	Y	N	N	--	4–6 weeks	Local: Cidofovir Systemic: Celecoxib
4	Best et al. [17]	United States	2017	10	M	Juvenile	Y	Y	Y	--	4 weeks	Local: Cidofovir Systemic: Interferon alpha, indole 3 carbinol, celecoxib
5	Best et al. [17]	United States	2017	18	M	Juvenile	Y	Y	Y	--	6 weeks	Local: Cidofovir, Bevacizumab Systemic: Interferon, leflunomide
6	Best et al. [17]	United States	2017	21	M	Juvenile	Y	Y	Y	--	6 weeks	Local: Cidofovir
7	Best et al. [17]	United States	2017	86	M	Adult	N	N	Y	--	3 months	Local: Cidofovir
8	Best et al. [17]	United States	2017	62	M	Adult	N	Y	Y	--	12 months	None
9	Cuestas et al. [20]	Argentina	2019	6	M	Juvenile	N	Y	Y	2	4 weeks	None
10	Nagel et al. [21]	Germany	2009	32	M	Juvenile	N	Y	N	4 per year during 12 years	8 weeks	Local: Interferon, Cidofovir
11	Gates et al. [22]	United States	2020	0.583333	M	Juvenile	Y	Y	N	--	10 months	None
12	Zur et al. [23]	United States	2016	12	F	Juvenile	Y	Y	Y	>500	1–4 weeks	Local: Cidofovir, interferon alpha Systemic: Celecoxib, Propanolol, Azythormycin, Gardasil
13	Mhor et al. [16]	Germany	2014	43	M	Juvenile	N	Y	Y	>30	--	Multiple laser ablations
14	Mhor et al. [16]	Germany	2014	49	M	Adult	Y	N	N	2	6 months	2 laser ablation
15	Mhor et al. [16]	Germany	2014	56	F	Adult	Y	N	N	16	--	16 Laser ablations
16	Mhor et al. [16]	Germany	2014	8	F	Juvenile	Y	Y	Y	>30	--	Multiple laser ablations Local: interferon-alpha, cidofovir and celecoxib
17	Mhor et al. [16]	Germany	2014	34	M	Adult	Y	N	N	6	--	laser vaporization and radical surgery via midfacial degloving, radiotherapy,
18	Fernandez-Bussy et al. [24]	Chile	2017	42	M	Adult	N	Y	N	3	--	Local: ablation with argon plasma coagulation
19	Carnevale et al. [14]	Spain	2018	5	M	Juvenile	Y	Y	Y	10	--	Multiples surgeries and laser ablations Local: Intepheron alpha, Cidofovir Systemic: Gardasil
20	Carnevale et al. [14]	Spain	2018	9	F	Juvenile	Y	Y	N	47	--	Mulple surgeries
21	Hamdi et al. [25]	United States	2020	12	M	Juvenile	Y	Y	Y	54	--	Multiples surgeries and laser ablations Local: Intepheron alpha, Cidofovir Systemic: Indole 3 Carbinol
22	Hamdi et al. [25]	United States	2020	6	M	Juvenile	Y	N	N	34	-	Local: Cidofovir
23	Bedoya et al. [15]	United States	2017	87	M	Adult	N	N	Y	5	--	Local: Cidofovir, intralesional stent Systemic: Gardasil
24	Bedoya et al. [15]	United States	2017	63	M	Adult	Y	Y	Y	>6	--	--
25	Evers et al. [26]	United States	2020	Betweenn 8 and 56	Both	Juvenile and adult	Y	Y	Y	>10	--	Lobe resection, surgical interventions, laser ablations, radiochemotherapy
27	Ortiz et al. [27]	Colombia	2021	Betweenn 1 and 8	Both	Juvenile	Y	Y	Y	>10	--	Debridement procedures
28	Gorelik et al. [28]	United States	2021	5	F	Juvenile	Y	N	N	1	--	Debridement procedures
29	Goyal et al. [29]	India	2021	4 and 24	F	Juvenile	Y	N	N	>10	--	Local Bevacizumab

**Table 2 children-10-00054-t002:** Systemic bevacizumab treatment, dosage and outcomes in patients with RRP. CR, Complete Regression, PR, Partial Regression, VGPR, very good partial remission, NA Not Applicable.

Case	Author	Country	Year	Prior Therapy	Dose (mg/kg)	Inicial Dosing Interval (q Week)	Duration of Treatment (Months)	Stable Dosing Interval (q Week)	Total Cycles Received	Response	Pulmonary Response	Surgical Interval Post Bevacizumab	Side Events
1	Best et al. [17]	United States	2017	Local: Cidofovir, bevacizumab, photodynamic therapy. Systemic. Interferon alpha	10	3	12	8	--	PR	Not known	q3 months	Treatment changed secondary to diagnosis of malignancy
2	Best et al. [17]	United States	2017	Local: Cidofovir Systemic: Interferon alpha, propanolol, celecoxib, Gardasil	10	4	21	8	--	PR	CR	q4 months	Proteinuria
3	Best et al. [17]	United States	2017	Local: Cidofovir Systemic: Celecoxib	10	2	10	12	--	CR	.	No surgery required	None
4	Best et al. [17]	United States	2017	Local: Cidofovir Systemic: Interferon alpha, indole 3 carbinol, celecoxib	5	3	18	3	--	PR	Stable	q6 months	None
5	Best et al. [17]	United States	2017	Local: Cidofovir, Bevacizumab Systemic: Interferon, leflunomide	10	3	7	6	--	PR	Not known	q6 week with no excision needed	Hemoptysis
6	Best et al. [17]	United States	2017	Local: Cidofovir	10	3	5	12	--	PR	Not known	q3 months	None
7	Best et al. [17]	United States	2017	Local: Cidofovir	10	3	10	8	--	PR	PR	No surgery required	None
8	Best et al. [17]	United States	2017	None	10	2	8	6	--	PR	PR	No surgery required	None
9	Cuestas et al. [20]	Argentina	2019	None	10	4	6	12	--	PR	PR	No surgery required	None
10	Nagel et al. [21]	Germany	2009	Local: Interferon, Cidofovir	10	3	6	10	--	CR	CR	No surgery required	None
11	Gates et al. [22]	United States	2020	None	5 and then 10	4	12	.	--	PR	-	No surgery required	None
12	Zur et al. [23]	United States	2016	Local: Cidofovir, interferon alpha Systemic: Celecoxib, Propanolol, Azythormycin, Gardasil	10	4–5	15.5	14	-	CR	CR	Tracheal debridement at 12 months after bevacizumab starting. Laryngeal debribement at 15 months after bevacizumab starting.	Mild proteinuria
13	Mhor et al. [16]	Germany	2014	Multiple laser ablations	10	2	27	16	16	VGPR	PR	No surgery required	Hypertension
14	Mhor et al. [16]	Germany	2014	2 laser ablation	10	3	2	.	3	PR	NA	Laryngectomy for malignant transformation	None
15	Mhor et al. [16]	Germany	2014	16 Laser ablations	10	--	.	.	6	VGPR	NA	One infraglottic lesion, which had not responded to the same extent was treated with additional laser therapy a los 14 días	None
16	Mhor et al. [16]	Germany	2014	Multiple laser ablations Local: interferon-alpha, cidofovir and celecoxib	5	2	2	12	9	VGPR	CR	One surgery due to granulloma at the distal end of the tracheal cannula	None
17	Mhor et al. [16]	Germany	2014	laser vaporization and radical surgery via midfacial degloving, radiotherapy,	15	3	2	.	6	CR	NA	No surgery required	None
18	Fernandez-Bussy et al. [24]	Chile	2017	Local: ablation with argon plasma coagulation	5 and then 10	3	4	12	6	CR	NA	No surgery required	None
19	Carnevale et al. [14]	Spain	2018	Multiples surgeries and laser ablations Local: Intepheron alpha, Cidofovir Systemic: Gardasil	10	4	46	10	--	CR	CR	Two surgical procedures due to laryngeal lesions	Mild proteinuria
20	Carnevale et al. [14]	Spain	2018	Mulple surgeries	5 and then 7.5	3	6	3	8	CR	NA	No surgery required	None
21	Hamdi et al. [25]	United States	2020	Multiples surgeries and laser ablations Local: Intepheron alpha, Cidofovir Systemic: Indole 3 Carbinol	10	3	54	3	15	CR	PR	Seven microdebriders	None
22	Hamdi et al. [25]	United States	2020	Local: Cidofovir	10	1	.	.	1	CR	CR	One surgery needed	None
23	Bedoya et al. [15]	United States	2017	Local: Cidofovir, intralesional stent Systemic: Gardasil	5	2	--	6	-	VGPR	VGPR	No surgery required	Hemoptysis
24	Bedoya et al. [15]	United States	2017	--	10	2	--	3	--	VGPR	VGPR	No surgery required	Mild Increase of basal hypertension
25	Evers et al. [26]	United States	2020	middle lobe resection with lymphadenectomy	10	2	--	3	68	VGPR	VGPR	51 months	Proteinuria, Hypertension
26	Evers et al. [26]	United States	2020	--	10	3	--	3	3	PR	NA	laser ablation	None
27	Evers et al. [26]	United States	2020	Laser ablations	10	10	--	--	34	CR	NA	17 months	Proteinuria, Hypertension
28	Evers et al. [26]	United States	2020	Laser ablations	5	2	--	--	33	CR	VGPR	32 months	None
29	Evers et al. [26]	United States	2020	--	15	3	--	--	15	PR	NA	--	None
30	Ortiz et al. [27]	Colombia	2021	Tracheostomy and surgical debridement	10	3	--	--	3	VGPR	NA	--	None
31	Ortiz et al. [27]	Colombia	2021	Tracheostomy	10	3	--	--	3	CR	NA	--	None
32	Ortiz et al. [27]	Colombia	2021	Debridement	10	3	--	--	4	CR	NA	--	None
33	Gorelik et al. [28]	United States	2021	Surgical debridement, Intralesional Bevacizumab	10	3	--	--	8	CR	NA	--	None
34	Goyal et al. [29]	India	2021	three doses of HPV vaccine Cervarix	10	3	--	--	--	VGPR	NA	--	None

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
