# Peer review of "Systemic Bevacizumab for Recurrent Respiratory Papillomatosis: A Scoping Review from 2009 to 2022"

_children, 2022, doi:10.3390/children10010054_

Round 1

Reviewer 1 Report

Could the authors comment regarding the efficacy if any of vaccination against HPV of the parents or siblings

Except for surgery and the investigational treatment did any of the children presented receive any other treatment including antibiotics?

How often were the presented cases follow-up

Could the author comment regarding the life  span and also the quality of life of the patients

Author Response

Thank you very much for your corrections to the manuscript. 
We have again revised the manuscript in which we described a case report and a systematic review.
when reviewing the case report, we realized the inconsistencies and errors, so we decided to eliminate the case report and improve the systematic review.
Together with the elimination of the clinical case, it was decided to eliminate some authors Paola Perez, Jaume Patiño, Diego Medina and Harry Pachajoa since their major contribution was with the clinical case, so they are excluded and two new authors Estephania Candelo and Maria Guerra are included. 
To improve the systematic review we searched for records identified through ClinicalTrials, EMBASE, MEDLINE (PubMed), LILACS, and Global Health (Ovid), Scopus.

Reviewer 2 Report

The manuscript with ID: children-1858225, titled “Systemic Bevacizumab for Recurrent respiratory papillomatosis. Case report and scoping review”, deals with a rare and severe disease. Reported data and the literature review are interesting and stimulating. Nonetheless the manuscript needs a number of improvements to be adequate for publication.

Introduction:

This section is somewhat approximate and incomplete, and the following points must be clarified/amended:

Acronyms must be introduced the first time any unusual definition is used.

Please introduce the acronyms “JO-RRP” and “AO-RRP” in line 39.

Line 42-43: “There is no cure for systemic papillomatosis, and no treatment modality has been shown to be effective”. This sentence is overstated. The expression “not resolutive” would be more appropriated. 

Line 44: as above the expression “Development of extra laryngeal papillomatosis has been seen in 30% of cases,” should be reformulated in a more cautious form, for instance: “…has been seen in up to 30% of cases”.

Line 46. The sentence “In addition, RRP is associated with an increased risk of malignancy” is too generic and  alarming considering that expected rate of malignant progression is rather low, although not negligible. Please give a numerical estimation of malignant progression and add a reference.

LINES 49-50.  “other risk factor(s) include Serotype 11 and the presence of tracheostomy”. Risk for what?? Please be more detailed and precise and reformulate this sentence.”

Please note that HPV11 is a GENOTYPE!! Please correct.

The reference number 7 is a case report including no experimental data about risk factors and therefore is not adequate here.

Lines 58 -61 Antivirals are mentioned.  A review is expected to give  a wide panorama of a working field. Have other biological modifiers or monoclonal Abs been proposed or used in RRP? whether yes  or not this point should be openly stated in the text 

Lines 65-71. The theoretical and experimental reasons suggesting the use of Bevacizumab in RRP should be at least listed and briefly discussed as well as its potential toxic or unwanted effects.

Clinical case.

Line 113 – 120: please report  and briefly discuss the base level of peripheral IgG; IgA and IgM, the IgG response to HBV vaccine and the result of PHA stimulation.

RRP is a rare condition of interest to a large number of different specialists with widely different knowledge ed expertise: please explain and briefly discuss what is the FCGR3A gene, what its functions, physiological roles and how does it impact on HPV6/11 infection and disease.

Lines 125 – 129 How and to what level was the IgG deficit corrected?

At Line 138 please add “Scoping review” at Results as a subheading.

Line 139 - 140: The manuscript reads “In this research strategy, 175 unique records were identified. Fifteen studies were eligible to be included in this scoping study. After applying inclusion and exclusion criteria, fourteen studies were included.” while in figure 1 (at line 182) a total 176 studies are mentioned, 11 are reported as eligible, 1 as not eligible with reasons and 10  were included in qualitative Synthesis. Please correct and specify why these discrepancies.

Line 148.         Please report fraction of years as number of twelfth (e.g.: 7/12) instead of decimal fractions (e.g.: 0.58).

Line 156. The manuscript reads “The median of the reported interventions was 13 (min 3; max 500)”. Please specify that it refers to the median number of surgical procedures patients have undergone.

Table 1 and table 2 lack a legend of symbols used

In table 1 the first column is titled study but it seems should be “patient”. Why just 29 are included?

Line 168. “After the systemic bevacizumab scheme, 12 (35%) patients did not require surgery again due to RRP” . How long were they monitored after the Bevacizumab withdrawal?

Discussion

Line 237. Once again the reference no 7 is inappropriate here.

Line241. Where the 24 cases of RRP patients treated with Bevacizumab came up?

 by each patient?

Lines 259 – 265. A review is expected to be read by a large number of professional with largely divergent knowledges ed expertise. Please specify what are L66H and FCGR3A, what their potential pathological relevance for RRP and the reason for including them in the discussion?

How long paediatric and adult patients were followed up after Bevacizumab suspension?

What are the potentials toxic effect of bevacizumab in rapidly growing organisms such as those of paediatric patients? Please add a short paragraph and discuss breifly

References

Although not necessary, including the doi number in any bibliography item would greatly improve the manuscript value.

As a matter of fact I found a number of difficult points  and of potential mistakes throughout the manuscript. Thus, although I am not a native English speaking person I would advise to have the manuscript revised by a professional English speaking  writer

Author Response

Dear Reviewer 2

Thank you very much for your corrections to the manuscript.

We have again revised the manuscript in which we described a case report and a systematic review.

when reviewing the case report, we realized the inconsistencies and errors, so we decided to eliminate the case report and improve the systematic review.

Together with the elimination of the clinical case, it was decided to eliminate some authors Paola Perez, Jaume Patiño, Diego Medina and Harry Pachajoa since their major contribution was with the clinical case, so they are excluded and two new authors Estephania Candelo and Maria Guerra are included.

To improve the systematic review we searched for records identified through ClinicalTrials, EMBASE, MEDLINE (PubMed), LILACS, and Global Health (Ovid), Scopus.

Introduction

Acronyms are introduced the first time the term is mentioned.

Line 42-43 The sentence was reworded and the term was replaced by "There is no resolutive modality".

Line 44. it was reformulated in a more cautious form "Development of extra laryngeal papillomatosis has been seen up to 30% of cases".

Line 46. it was reformulated as follows: RRP is associated with a risk of malignancy of RR: 8.0 (CI 95%: 1.1-60.3) , references : Fortes HR, von Ranke FM, Escuissato DL, Araujo Neto CA, Zanetti G, Hochhegger B, et al. Recurrent respiratory papillomatosis: A state-of-the-art review. Vol. 126, Respiratory Medicine. W.B. Saunders Ltd; 2017. p. 116-21.

  1. Kashima H, Mounts P, Leventhal B, Hruban RH. Sites of predilection in recurrent respiratory papillomatosis. Annals of Otology, Rhinology & Laryngology. 1993;102(8):580-3.
  2. Lee LA, Cheng AJ, Fang TJ, Huang CG, Liao CT, Chang JTC, et al. High incidence of malignant transformation of laryngeal papilloma in Taiwan. Laryngoscope. 2008 Jan;118(1):50-5.

Line 49. Reworded the paragraph, correcting the word serotype to genotype.

Lines 58-61 made open discussion in the text regarding the use of antibirals, biologic modifiers or monoclonal antibodies.

Lines 65-71 listed and discussed the reasons for the use of Bevacizumab in RRP as well as its effects.

corrections regarding the clinical case do not apply given previously explained.

Reviewer 3 Report

This is an interesting pediatric clinical case report of recurrent respiratory papillomatosis. It is followed by a thorough review of the literature on the use and efficacy of bevacizumab for this disease.

However, the work needs careful review of English and of typos and punctuation errors.

Following are some suggestions to enhance the understanding of the paper:

Introduction

Line 39, shorten juvenile-onset (JO) as done for adult-onset form (AO).

Better specify the main sites of respiratory localization (laringeal and extra laryngeal sites).

In line 46, clarify whether “malignancy” refers to transformation of papillomas from benign to cancerous forms.

Describe the ways of transmission (e.g., intrapartum route in early-onset forms) and whether there are diseases associated with or predisposing to the development of RRP and RCP, such as primary immunodeficiencies or genetic causes.

Specify whether bevacizumab is used systemically only for RRP or also for RCP.

Results

Divide chapter 3-results-into two paragraphs (3.1 and 3.2) regarding the clinical case and the literature review.

Line 102, the child appears to be 6 months old while in the introduction it is described a 21-month-old child. Please clarify.

Line 109, specify whether the granulomas were pulmonary or intraperitoneal or associated (and/or?).

Specify which genetic test was performed, whether sanger sequencing or NGS or other.

Line 128, add a verb after the term "monthly".

In the conclusion of the case, describe how the patient had responded to the therapy before the family's decision to discontinue the treatment.

Remove the description of the child's lesions (“Image 1”) from the text (line 119-121) because it is already given as a figure caption and call it "Figure 1", citing it in the clinical case description. Then, order the number of figures that are cited in the text.

If possible, report in the review paragraph the mean age at first surgery before bevacizumab and the mean age at first bevacizumab administration.

Specify what is the meaning for “complete” or “partial” response.

Table 1

Does the “age at treatment start” refer to bevacizumab or to the first treatment in general performed before bevacizumab?

When the column is not completed but the hyphen is reported, does it mean that the data is not available? If so, please specify.

Table 2

Add a caption reporting the meaning of abbreviations.

Author Response

Thank you very much for the corrections to the manuscript. 
a new revision of the manuscript was made in which we described a case report and a systematic review.
when reviewing the case report we realized the inconsistencies and mistakes, so we decided to eliminate the clinical case and improve the systematic review.
Together with the elimination of the clinical case, it was decided to eliminate some authors Paola Perez, Jaume Patiño, Diego Medina and Harry Pachajoa since their major contribution was with the clinical case, so they are excluded and two new authors Estephania Candelo and Maria Guerra are included. 
To improve the systematic review we searched for records identified through ClinicalTrials, EMBASE, MEDLINE (PubMed), LILACS, and Global Health (Ovid), Scopus.

Introduction
In the introduction section, the corresponding modifications were made according to your comments, 

Table 1 
age at treatment start refers to the age at which management with bevacizumab was given.
When the column is not completed but the hyphen is reported it means that the data is not available.

Table 2 
a caption has been added to inform of the meaning of the abbreviations.
corrections regarding the clinical case do not apply given previously explained. 

Round 2

Reviewer 2 Report

The manuscript is linguistically puzzling and sometimes diverting or misleading and is therefore   hard to read. It has to be throughout revised by a professional writer.

The manuscript lacks lines number. Passages from the manuscript are here reported in brackets and “Italics”, my comments in regular characters. This makes difficult and time consuming the referee’s work.

A selection of critical points is following.

In the background section, just following the RRP, the Recurrent Cutaneous Papillomatosis is introduced and briefly discussed although it is not further discussed in any part of the manuscript nor is cited by any of the published study considered in this review.

The survey of literature provided in the background is approximate and superficial. For instance, in the passage “Management with antivirals is the first line of adjuvants, but their results are heterogeneous and, in some cases, are not effective (9)” authors should be more precise and detailed. Which antiviral resulted partially effective and which one not effective? In which condition were they tested? What the success/failure rates of different antivirals? Were they administered along the same route and at comparable dosage?

Another available treatment option is monoclonal antibodies like anti PDL-1 and Bevacizumab (10). Bevacizumab is a monoclonal antibody targeted against vascular endothelial growth factor (VEGF), inhibiting angiogenesis and tumor growth (11)” Is there any proposed rationale to choose  Bevacizumab among the very large number of antiproliferative/antineoplastic monoclonal Ab or it was just a serendipitous decision?

This drug has been used in sub-lesional areas of RCP and RRP as an adjuvant to surgical management” Did the authors mean “…delivered through peri-lesional administration?

Some authors have reported that high doses of sub-lesional Bevacizumab could increase the time interval between procedures (what procedures?) without adverse events, but there is variability in response and remission outcomes (12,13)” The content of these two reports should be briefly recalled and commented enabling the readers to achieve an independent opinion about pros and cons.

RRP is mostly a juvenile condition. Are the adverse effects in paediatric patients the same of adults?

Results.

The Figure 1 quoted in the text is instead named “1.prisma flow chart”  in the annexes. Please uniform the names.

The text reporting the results of table 1 should be sustained by plots and diagrams  which would greatly improve the readability of text.

The statement: “The median of the reported interventions was 13 (min 3; max 500)” is incomplete, please add the unit of measurement: did some patient underwent up to 500 surgical removals? Were they days of interval between surgeries? Weeks of interval?, Months of interval?

As above plots and diagrams would greatly improve the readability of results included in table 2 .

A stable dosing interval was reported in 30/34 patients. The median regular dosing  interval in these patients was 8 (min 3; max 8)”. What the unit? days?

Twenty patients (58,8%) were reported as complete responses. Of them, eight had excellent response. Twelve patients had partial response”. To me it seems that 12 Patients with partial responses were included in the 20 recorded as complete responses which is unacceptable of course!!

After the systemic bevacizumab scheme, 12 (35%) patients did not require surgery again due to RRP. Six required up to two additional interventions for punctual lesions that did not respond. One patient required seven microdebrider procedures. Seven patients continued to require surgical interventions, but the interval between these was extended from weeks to months. One patient required a laryngectomy due to malignant transformation”. 12+6+1+7+1 = 27. What about the remaining 7 patients?

Discussion.

“In this scoping review, we found 24 cases of patients with RRP treated with systemic Bevacizumab described in the literature, adding our patient, the youngest boy with RRP and cutaneous who has received systemic Bevacizumab. Systemic Bevacizumab can be a feasible adjuvant therapy for managing and controlling severe RRP patients”. The meaning of this passage is obscure to me. A number of 34 patients, is reported elsewhere in the text,  please correct.

Side effects

The only observed side effects in the pediatric population with systemic bevacizumab treatment were mild proteinuria in two patients, with no long-lasting side effects”. Did these adverse effects lead to treatment redrwawal/suspension/modification?

Author Response

dear reviewer,

the manuscript has undergone a linguistic and writing revision by a native professional, and a translation certificate is attached.
line numbering was performed
clarifications were also made in relation to the introduction, methods and results.
the units of measurement in the results were corrected and the redaction in the discussion.